# Assessing the Utility of Acoustic Radiation Force Impulse in the Evaluation of Non-Alcoholic Fatty Liver Disease with Severe Obesity or Steatosis

**DOI:** 10.3390/diagnostics14111083

**Published:** 2024-05-22

**Authors:** Yeo Wool Kang, Yang Hyun Baek, Jong Hoon Lee, Young Hoon Roh, Hee Jin Kwon, Sang Yi Moon, Min Kook Son, Jin Sook Jeong

**Affiliations:** 1Department of Internal Medicine, Dong-A University College of Medicine, 32 Daeshingongwonro, Seo-gu, Busan 49201, Republic of Korea; ywkang8756@dau.ac.kr (Y.W.K.); jh2002@dau.ac.kr (J.H.L.); sang4401@dau.ac.kr (S.Y.M.); 2Department of General Surgery, Dong-A University College of Medicine, 32 Daeshingongwonro, Seo-gu, Busan 49201, Republic of Korea; gsryh@dau.ac.kr; 3Department of Radiology, Dong-A University College of Medicine, 1,3-ga Dongdaesindong, Seo-gu, Busan 49201, Republic of Korea; risual@dau.ac.kr; 4Department of Physiology, Dong-A University College of Medicine, 32 Daeshingongwonro, Seo-gu, Busan 49201, Republic of Korea; physionet@dau.ac.kr; 5Department of Pathology, Dong-A University College of Medicine, 32 Daeshingongwonro, Seo-gu, Busan 49201, Republic of Korea; jsjung1@dau.ac.kr

**Keywords:** non-alcoholic fatty liver disease, acoustic radiation force impulse, non-invasive test, hepatic fibrosis

## Abstract

Background: Non-alcoholic fatty liver disease (NAFLD) encompasses a heterogeneous spectrum ranging from simple steatosis to fibrosis and cirrhosis. Fibrosis, associated with long-term overall mortality and liver-related events, requires evaluation. Traditionally, liver biopsy has been the gold standard for diagnosing fibrosis. However, its invasive nature, potential complications, and sampling variability limit widespread use. Consequently, various non-invasive tests have been developed as alternatives for diagnosing fibrosis in NAFLD patients. Aim: This study aimed to compare the accuracy of non-invasive tests (NITs) and evaluate the diagnostic accuracy of acoustic radiation force impulse (ARFI), one of the point shear wave techniques, compared to conventional methods, assessing its effective role in diagnosis. Methods: This is a retrospective study; a total of 136 patients diagnosed with fatty liver disease through ultrasonography were enrolled. The anthropometric data of the patients were collected on the day of admission and blood tests, measurements of ARFI, and a point shear test were conducted using abdominal ultrasound; a biopsy was performed the following day. In addition, we calculated the aspartate aminotransferase-to-platelet ratio index (APRI) index based on four factors (FIB-4) and the NAFLD fibrosis score (NFS). Subsequently, we assessed the diagnostic accuracy of NITs within various subgroups based on the extent of obesity, steatosis, or NAFLD activity score. Results: ARFI has been shown to have the highest diagnostic value among various NITs, with AUROC values of 0.832, 0.794, 0.767, and 0.696 for ARFI, APRI, FIB-4, and NFS, respectively. In the morbidly obese subgroup, the AUROC values of ARFI, APRI, FIB-4, and NFS were 0.805, 0.769, 0.736, and 0.674. In the group with severe steatosis or non-alcoholic steatohepatitis (NASH), the AUROC values were 0.679, 0.596, 0.661, and 0.612, respectively, for severe steatosis and 0.789, 0.696, 0.751, and 0.691, respectively, for NASH. Conclusions: In conclusion, ARFI is not affected by various factors and maintains diagnostic accuracy compared to serum NITs. Therefore, we can recommend ARFI as a valuable diagnostic test to screen for advanced fibrosis in patients with NAFLD.

## 1. Introduction

Non-alcoholic fatty liver disease (NAFLD) affects approximately one-quarter of the world’s population and causes liver-related complications [1]. Recent reports indicate a rising prevalence in Asian nations, such as Korea, with estimates ranging from 21% to 44% and a median of 30% in the general population [1]. NAFLD encompasses a histological spectrum, extending from simple steatosis to non-alcoholic steatohepatitis (NASH) and progressing to advanced fibrosis, ultimately resulting in cirrhosis. Studies have highlighted that advanced fibrosis and cirrhosis are pivotal predictors of mortality and liver-related complications [2]. Therefore, it is important to identify advanced fibrosis and cirrhosis in NAFLD patients. Conventional diagnostic methods for assessing liver fibrosis in NAFLD patients have traditionally relied on liver biopsy, which, while effective, is an invasive procedure associated with potential risks. These risks encompass not only minor complications like pain (20%) but also major complications, including haemobilia, abdominal bleeding (0.5%), and even rare instances of mortality (0.009–0.12%) [3,4]. Other challenges associated with liver biopsy include sampling error and inter- and intra-observer variability [5]. Recently, many studies have been conducted to evaluate liver fibrosis using non-invasive tests (NITs) to minimize liver biopsy in screening fibrosis. Numerous guidelines have suggested the use of algorithms such as index based on four factors (FIB-4), the NAFLD fibrosis score (NFS), transient elastography (TE), magnetic resonance elastography (MRE), and acoustic radiation force impulse (ARFI) for screening fibrosis and determining the necessity of liver biopsy in patients [1,6,7]. MRE utilizes a quantitative phase-contrast-based MRI technique to measure tissue stiffness, thereby indirectly assessing hepatic fibrosis. TE is currently the most commonly used imaging technique for evaluating liver fibrosis but it has limitations in cases of severe obesity or narrow intercostal spaces and requires the purchase of additional equipment for testing. Similarly, MRE is considered the most accurate non-invasive imaging technique for assessing liver fibrosis but like TE, it requires additional equipment and is costly, limiting its use to clinical research. Therefore, in this study, we utilized ARFI, a point shear wave imaging technique, as a non-invasive imaging method. We evaluated whether ARFI testing is useful for diagnosing progressive fibrosis compared to the currently used serological non-invasive tests and assessed whether diagnostic accuracy is maintained in patients with limitations for TE.

## 2. Methods

### 2.1. Study Population

From November 2017 to December 2022, 136 people who were suspected of having NAFLD and underwent a biopsy at Dong-A University Hospital (Busan, Republic of Korea) were retrospectively enrolled. The study’s inclusion criteria were as follows: (i) the presence of a bright echogenicity as observed in ultrasonography and (ii) an unexplained elevation of liver enzymes exceeding the upper reference limit. In this study, patients with (i) B or C viral hepatitis, (ii) autoimmune hepatitis, (iii) excessive alcohol consumption defined as >30 g/day (men) and >20 g/day (women), (iv) focal liver lesions other than simple cysts and hemangioma on ultrasound, and (v) those who did not undergo ARFI imaging tests were excluded (Figure 1).

### 2.2. Serologic Biomarkers for Estimation of Liver Fibrosis

Patient assessments were performed following admission, utilizing anthropometric data obtained on the day of admission. On the same day, after a 12-h fast, blood samples, liver biopsies, and ARFI were performed. The blood tests encompassed assessments for various factors, including AST, ALT, platelet count. fasting glucose, albumin, and HbA1c. The reference ranges are as follows: AST and ALT: 0–40 U/L, platelet count: 150–450 × 10^9^/L, albumin: 3.4–5.4 g/dL, fasting glucose: 74–100 mg/dL, and HbA1C: 5.7–6.4%.

The AST to platelet ratio index (APRI) was calculated using the following formula:$$[(AST\;level/upper\;limit\;of\;normal\;(ULN))/platelet\;count\;(10^{9}/L)]\times 100.\;$$

FIB-4 was calculated using the following formula:$$age\;(years)\times AST\;(U/L)/[platelet\;count\;(10^{9}/L)\times ALT{\left(U/L\right)}^{1/2}].\;$$

NFS was calculated using the following formula: $$-1.675+0.037\times age\;(years)+0.094\times BMI\;(kg/m^{2})+1.13\times IFG\;or\;diabetes\;(yes\;=\;1,\;no\;\;=\;0)+0.99\times AST/ALT\;ratio-0.013\times platelet\;count\;(\times 10^{9}/L)\;-0.66\times albumin\;(g/dL).$$

### 2.3. Imaging Test for Estimation of Liver Fibrosis—Acoustic Radiation Force Impulse (ARFI)

Point shear-wave elasticity technology uses ARFI to stimulate the liver tissue and to generate shear waves that propagate into the liver. The shear-wave velocity increases with the severity of fibrosis. ARFI was performed using the Philips EPIQ ultrasound equipment (Bothell, WA, USA). Two experienced radiologists, who were blinded to the clinical and biochemical data, conducted the examination. They utilized software 6.3.7.745 version to measure liver stiffness measurements (LSMs). Prior to ARFI imaging, traditional ultrasonography was used to assess gross morphology and LSM was measured by placing a cursor over a 10 × 5 mm region-of-interest within the liver parenchyma (at least 3 cm below the Gleason capsule, an area free of focal lesions or blood vessels). Ten valid measurements were obtained from patients and the median shear wave velocity value was considered to represent liver stiffness in meters per second (m/s) (Figure 2).

### 2.4. Liver Histopathology

An experienced pathologist with over 20 years of expertise in liver histopathology evaluated and scored all liver biopsy specimens. Pathologists can obtain all the information about the patient from the tissue sample being interpreted, including biochemical and biological information, as well as underlying diseases. However, this information is unlikely to affect the biopsy results and it does not constitute a standard pathological diagnosis. Liver histopathology results were assessed using the NASH Clinical Research Network (CRN) staging system. The CRN staging system evaluates the degree of steatosis, lobular inflammation, cytological ballooning changes, and fibrosis. The NAFLD Activity Score (NAS) can be obtained as the sum of each score for steatosis, lobular inflammation, and cytologic ballooning. An NAS value of 5 or higher means NASH. Steatosis was graded according to the following criteria: <5% = 0, 5–33% = 1, 33–66% = 2, and >66% = 3. Fibrosis was assessed using the following criteria: no fibrosis = F0, mild to moderate zone 3 periarticular fibrosis or portal/periportal fibrosis = F1, zone 3 perisinusoidal and portal/periportal fibrosis = F2, bridging fibrosis = F3, and cirrhosis = F4. Steatosis was classified as minimal to mild if it was less than 33%, moderate if it ranged from 33 to 66%, and severe if it exceeded 66%. Significant fibrosis was defined as F2 or higher, while advanced fibrosis was defined as F3 or higher.

### 2.5. Ethics Committee Approval

Ethical approval was granted by the Dong-A Medical School Ethics and Medical Research Committee (DAUHIRB-17-197). Prior to enrollment, written informed consent was obtained from each research subject. All patients were enrolled in the Division of Gastroenterology and Hepatology at Dong-A University Hospital.

### 2.6. Statistical Analysis

Statistical analysis was performed using IBM SPSS Statistics version 22.0 (IBM Corp., Armonk, NY, USA) and MedCalc Software for Windows version 17.1 (MedCalc Software Ltd., 2022, Ostend, Belgium). Continuous variables were expressed as mean values with standard deviations and categorical variables were expressed as numbers with a percentage (%). The diagnostic performances of ARFI, APRI, FIB-4, and NFS were assessed by calculating the area under the receiver operating characteristic curve (AUROC) values and 95% confidence intervals (CI). For each test, the best cut-off value was associated with the Youden index. The AUROC values of the ARFI, APRI, FIB-4, and NFS for advanced fibrosis were compared using the DeLong test. *p*-values less than 0.05 were considered statistically significant.

## 3. Results

### 3.1. Baseline of Clinical and Histopathological Characteristics

Between January 2017 and December 2022, 136 people with a clinical suspicion of NAFLD who had undergone liver biopsy at Dong-A University Hospital were retrospectively analyzed. The baseline characteristics are presented in Table 1. Of the 136 patients, 71 (52.2%) were male and 65 (47.8%) were female. Furthermore, 76 (55.9%) patients had hypertension, 45 (33.1%) had IFG, and 48 (35.3%) had diabetes. A BMI of <23 kg/m^2^ was defined as non-obesity and a BMI of ≥30 kg/m^2^ was defined as morbid obesity in the 2020 Korean Society for the Study of Obesity guidelines [8]. The mean BMI was 29.83 kg/m^2^; only 3.7% of patients in this study were not obese, while 42.6% were morbidly obese. Thirty (22.1%) patients had advanced fibrosis, and seven (5.1%) patients had cirrhosis. In total, 44 patients (32.4%) had moderate steatosis and 54 patients (39.7%) had severe steatosis. Regarding NAS, 87 of the participants (63.9%) had NASH.

### 3.2. Diagnostic Accuracy of ARFI and Serologic Markers for Advanced Fibrosis

Among the NITs, ARFI, APRI, FIB-4, and NFS demonstrated diagnostic accuracy with respective AUROC values of 0.832 (95% CI, 0.751 to 0.914), 0.794 (95% CI, 0.711 to 0.878), 0.767 (95% CI, 0.656 to 0.877), and 0.696 (95% CI, 0.579 to 0.814). In our study’s AUROC analysis, the recommended cut-off values were >1.32, >0.637, >1.75, and >−0.3 for ARFI, APRI, FIB-4, and NFS, respectively. The sensitivity, specificity, positive predictive value (PPV), and negative predictive value (NPV) of ARFI, APRI, FIB-4, and NFS in line with these cut-off values are summarized in Table 2. The AUROC value of ARFI was superior to that of other tests and both ARFI and APRI demonstrated superior sensitivity and NPV compared to other non-invasive tests. In a comparative analysis assessing the statistical superiority of ARFI and APRI over the other tests, we found no statistically significant difference for APRI. However, it is worth noting that the diagnostic accuracy of ARFI was statistically superior to NFS (*p* = 0.0173), as depicted in Figure 3.

### 3.3. Diagnostic Predictive Accuracy of ARFI and Serologic Markers According to Subgroups, Such as Obesity, Steatosis, and NASH

We evaluated whether the diagnostic accuracy of NITs was influenced by the degree of obesity, presence of steatosis, and NASH. The values for each subgroup were as follows. In the analysis of diagnostic accuracy according to the degree of obesity, the AUROC of ARFI was 0.805 (95% CI, 0.682 to 0.929, *p* = 0.000) in the morbidly obese group, which indicated superior diagnostic accuracy compared with other non-invasive tests (Figure 4(1)). A comparative analysis of AUROC between each non-invasive test was conducted to determine whether ARFI was statistically superior to serological markers in the morbidly obese group. However, the results did not show statistical significance for ARFI over serologic markers such as APRI, FIB-4, or NFS (*p* = 0.6157, *p* = 0.4392, *p* = 0.1702).

The comparison of diagnostic accuracy of NITs for advanced fibrosis in the degree of steatosis subgroups is shown in Figure 4(2). In the severe steatosis group, the AUROC of ARFI, APRI, FIB-4, and NFS were 0.679 (95% CI, 0.519–0.840); 0.596 (95% CI, 0.427–0.765); 0.661 (95% CI, 0.481–0.841); and 0.612 (95% CI, 0.435–0.790), respectively. In the AUROC analysis of APRI, FIB-4, and NFS (except for ARFI), the 95% CI included a value of 0.5 or less and the *p*-value was higher than 0.05; therefore, the AUROC was not statistically significant in APRI, FIB-4, and NFS. Regardless of the degree of steatosis, only ARFI maintained a constant statistical significance.

When subgroups were divided using NAS, the AUROC values of ARFI, APRI, FIB-4, and NFS in the NASH group were 0.789 (95% CI, 0.687–0.891); 0.696 (95% CI, 0.580–0.811); 0.751 (95% CI, 0.632–0.870); and 0.691 (95% CI, 0.564–0.819), respectively (Figure 4(3). The diagnostic predictive accuracy of the ARFI was the best in this subgroup. In the NASH subgroup, the AUROC was compared and analyzed between NITs to determine whether the diagnostic predictive value of ARFI was statistically superior to those of other tests. The results represented that ARFI was not statistically superior to APRI, FIB-4, or NFS (*p* = 0.1502, *p* = 0.5330, *p* = 0.1379, respectively).

## 4. Discussion

Accurate evaluation of advanced fibrosis is critical for decisions and for predicting outcomes such as the occurrence of liver-related complications, the need for liver transplantation, and overall survival. While liver biopsy was traditionally the gold standard for assessing fibrosis, recent research has focused on non-invasive tests to reduce the potential risks associated with liver biopsy procedures. Based on these studies, diagnostic algorithms, including non-invasive tests, have been presented in various guidelines [1,7,9]. In our study, among non-invasive tests that predict fibrosis, APRI, FIB-4, and NFS were used as serological fibrosis markers and ARFI was used as an imaging method.

Originally, APRI was developed for patients with HCV or HIV/HCV coinfection [10]. In many cases, the thresholds were adjusted based on test results initially obtained from HCV patients and tailored for the assessment of advanced fibrosis in the context of NAFLD. In the analysis of 315 patients with NAFLD at Seoul National University, the AUROC value of APRI was reported as 0.830 (95% CI, 0.770–0.890; cut-off = 0.562; PPV = 38.1%, NPV = 94.9%; *p* = 0.374) [11]. In our study, the results of the AUROC values of APRI were similar to those of previous studies. APRI demonstrated a high NPV but a limited PPV, indicating its potential for excluding patients without advanced fibrosis, thereby helping to avoid unnecessary liver biopsies. Nonetheless, it exhibits reduced accuracy in detecting early fibrosis stages and can yield variable results due to fluctuations in ALT levels.

FIB-4 is a widely validated NIT and its simplicity makes it a useful tool to identify advanced fibrosis [12]. FIB-4 is recognized for its good diagnostic performance, consistently achieving AUROC values of 0.8 or higher in various studies assessing its efficacy in predicting advanced fibrosis [13]. Nonetheless, in older patients, FIB-4 exhibits decreased specificity, prompting the proposal of new thresholds for those aged 65 years and older [14]. In our study; the AUROC value was 0.767 (95% CI, 0.656–0.877), showing a relatively low diagnostic performance. FIB-4 varies greatly in value depending on age and the influence of platelet count because the FIB-4 formula includes age, platelet count, AST, and ALT. When analyzing the age of patients included in our study, the mean age was 44.9. Overall, 36.0% of the patients were younger than 35 years or older than 65 years of age. In other studies, patients with normal platelet counts were included and the age of the participants was limited to the range that could be evaluated; however, additional reasons for the low diagnostic performance in our study are needed.

In a study conducted in the United States, the NFS had an AUROC of 0.82 (95% CI, 0.76–0.88) for detecting advanced fibrosis [15]. In addition, in many published studies, the AUROC value was 0.8 or higher, which indicated a good diagnostic performance [16,17,18]. In our study, NFS had a relatively low AUROC value and showed a fair diagnostic predictive performance. NFS varies in value depending on whether it is with or without IFG and DM [19]. In the aforementioned studies, IFG or diabetes accounted for less than half of all patients. In our study, patients with IFG and diabetes accounted for 68.4% of the participants. The reason the AUROC value for NFS was low in our study may be because of the high proportion of patients with IFG and diabetes. Furthermore, the variations observed in the diagnostic performance of NFS among different studies may be attributed to the discrepancies in obesity prevalence between Asian and Caucasian populations affected by NAFLD [1,20]. In addition, studies have shown that sex, waist circumference, serum LDL-C, glomerular filtration rate, smoking, and exercise are factors that affect NFS [19]. The fact that the diagnosis of NFS was influenced by several factors suggests that its diagnostic value might vary depending on the characteristics of the population.

Although ARFI was used in our study, TE is more commonly used to screen for liver fibrosis. In a large study involving 246 NAFLD patients, TE demonstrated an AUROC of 0.93 (95% CI, 0.89–0.96) in the detection of advanced fibrosis [21]. Nevertheless, among overweight and obese patients, approximately 24–35% of the measurements were categorized as unreliable. The technique’s reliability is influenced by factors such as BMI, metabolic syndrome, and gender. While obesity poses certain limitations, this issue has been partially addressed through the introduction of a larger XL probe specifically designed to assess the liver at depths greater than 2.5 cm from the skin [22]. However, it is important to note that the validity and the establishment of cut-off values for this probe in assessing fibrosis have yet to be determined.

Compared with TE, ARFI offers a significant advantage as it can be seamlessly integrated into standard ultrasound equipment. Consequently, it can serve as a valuable complement to the traditional B-mode evaluation of the entire liver, particularly with improved reproducibility in obese and ascitic patients [23]. Recent data suggest that ARFI may have a diagnostic accuracy comparable to that of TE [24,25]. In contrast to TE, ARFI exhibits comparable accuracy with a notable advantage of lower rates of measurement failures. In our study, the AUROC value of ARFI was 0.832 (95% CI, 0.751–0.914), which is consistent with those of other published studies. However, ARFI has limitations, including the fact that it requires the operator to define the region of interest and obtain a series of LSMs. The values may vary depending on the skill level of the operator. Additionally, the good diagnostic performance of ARFI has been reported in a few studies and each study had a different cut-off; therefore, additional data on its optimal cut-off value and quality criteria are needed.

We categorized NAFLD patients into subgroups based on obesity, steatosis, and NASH. Notably, no prior research has conducted a comparative analysis of the diagnostic efficacy of non-invasive tests within these subgroups. Previous studies have shown that the degree of obesity, steatosis, or inflammation reduces the diagnostic accuracy of the NITs [26,27,28]. In our study, the proportion of patients with morbid obesity, severe steatosis, and NASH was 42.6%, 39.7%, and 63.9%, respectively. Our subgroup tended to have poor diagnostic accuracy for all non-invasive tests and ARFI consistently showed better diagnostic accuracy than the other non-invasive tests and was only statistically significant in patients with severe steatosis. Moreover, ARFI maintained good diagnostic accuracy while maintaining statistical significance with a *p*-value < 0.05 in each subgroup analysis.

This study comes with certain limitations. Firstly, there may be an antecedent bias due to the single-institution nature of the study and the inclusion of patients who underwent liver biopsy as deemed necessary. Future research might benefit from larger multicenter settings. Secondly, we did not perform TE, the most commonly used method for LSM, making a comparison between TE and ARFI unviable. Moreover, the range of cutoff values for classifying each stage of fibrosis in ARFI is relatively narrow, which might limit its practicality for classifying fibrotic stages. Lastly, during the ARFI test, only one radiologist conducted the evaluation, preventing another radiologist from observing and assessing it. Consequently, we could not assess the inter- and intra-observer variability of the results.

NAFLD is a diverse condition influenced by a range of factors and often co-occurs with various metabolic diseases; therefore, it is very difficult to find a suitable non-invasive test considering all influencing factors. NAFLD should be tested to show persistent diagnostic accuracy, regardless of various comorbid diseases or confounding factors. In this study, ARFI did not show statistical significance compared with APRI, FIB-4, and NFS when comparing the diagnostic accuracy of all subgroups but maintained good diagnostic values in various patient groups. In conclusion, our findings demonstrated the usefulness of ARFI as a screening test in patients with NAFLD, regardless of morbid obesity, NASH, or severe steatosis.

## 5. Conclusions

Compared to serum NITs, ARFI maintained diagnostic accuracy without being affected by various factors. Therefore, we propose that ARFI is a valuable diagnostic test, non-inferior to serum NITs, for screening advanced fibrosis in patients with NAFLD.

## Figures and Tables

**Figure 1 diagnostics-14-01083-f001:**
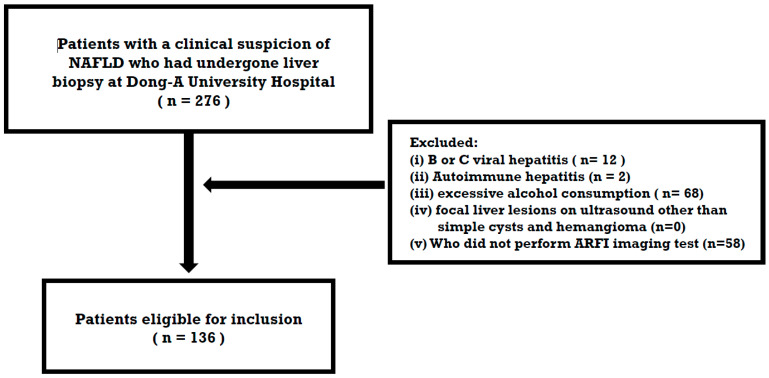
The flowchart of the study population.

**Figure 2 diagnostics-14-01083-f002:**
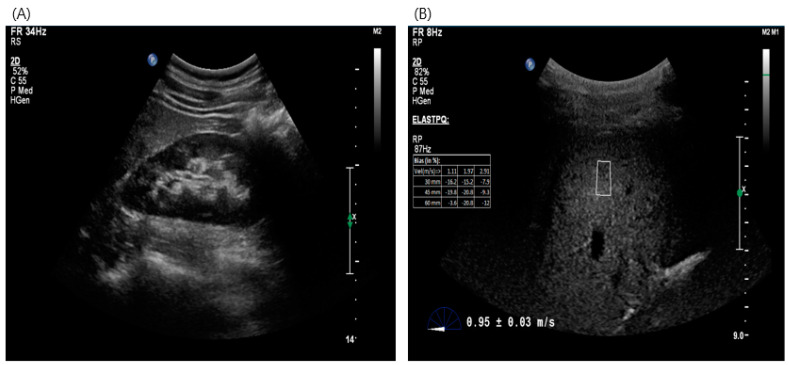
ARFI imaging. Simple abdominal ultrasound findings in an emergency patient (**A**) and ARFI test image (**B**).

**Figure 3 diagnostics-14-01083-f003:**
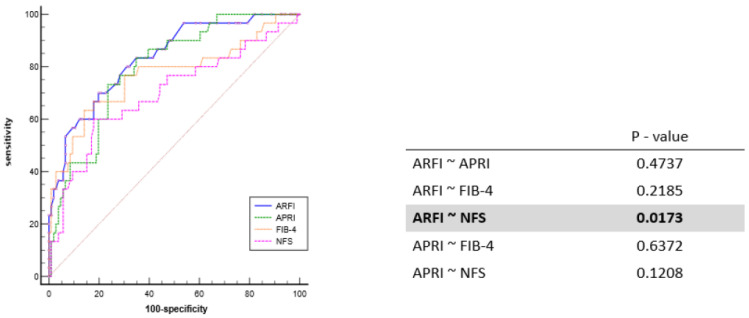
Comparison of diagnostic predictive value between each non-invasive fibrosis test.

**Figure 4 diagnostics-14-01083-f004:**
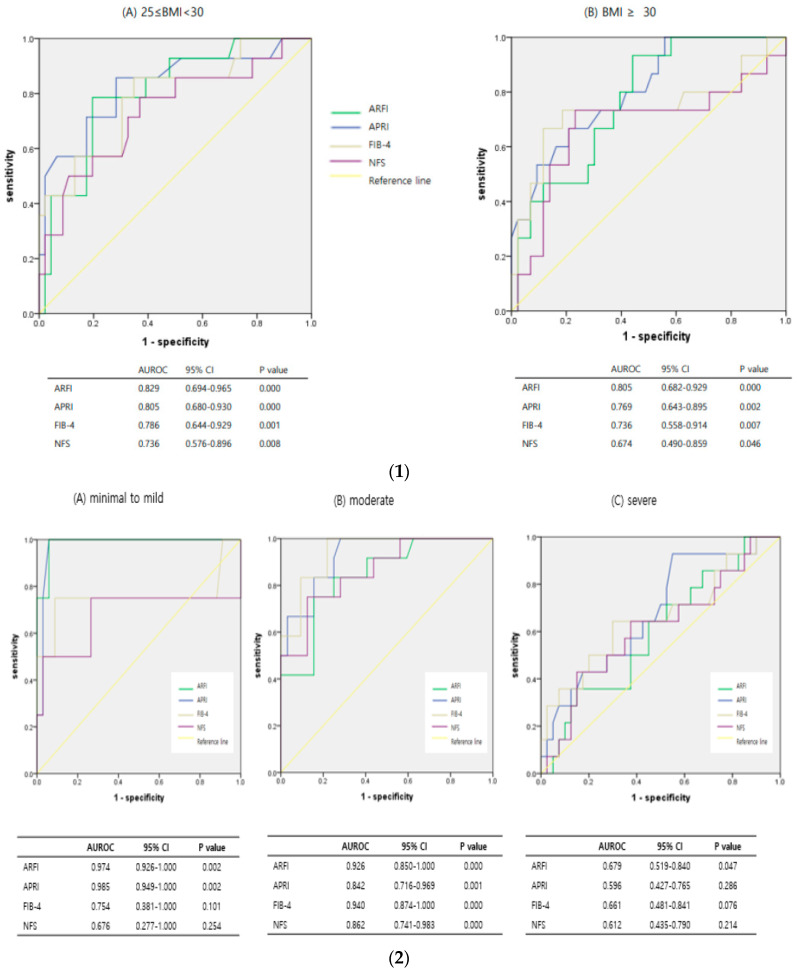
Diagnostic predictive value of each non-invasive test according to subgroups. (**1**) Diagnostic predictive value of each test with or without morbid obesity. (**2**) Diagnostic predictive value of each test according to the steatosis severity. (**3**) Diagnostic predictive value of each test with or without NASH.

**Table 1 diagnostics-14-01083-t001:** Baseline characteristics of the study population.

		Total (*n* = 136)
Demographic		
Age at biopsy, y, mean ± SD		44.92 (±14.34)
Male, *n* (%)		71 (52.2%)
Female, *n* (%)		65 (47.8%)
BMI, kg/m^2^, mean ± SD		29.83 (±4.38)
BMI ≥ 30		58 (42.6%)
Impaired fasting glucose, *n* (%)		45 (33.1%)
Diabetes, *n* (%)		48 (35.3%)
Hypertension, *n* (%)		76 (55.9%)
Biochemical profile (mean, (min, max))		
AST, U/L		59.92, (14, 345)
ALT, U/L		78.17, (9, 451)
Platelet count, 10^3^/uL		257.17, (114–642)
Albumin, g/dL		4.47, (3.2–5.3)
Fasting glucose, mg/dL		123.15, (68–333)
Histology		
Steatosis, *n* (%)	0	5 (3.7%)
	1	33 (24.3%)
	2	44 (32.4%)
	3	54 (39.7%)
Lobular inflammation, *n* (%)	0	22 (16.2%)
	1	31 (22.8%)
	2	50 (36.8%)
	3	33 (24.3%)
Ballooning, *n* (%)	0	45 (33.1%)
	1	55 (40.4%)
	2	36 (26.5%)
Fibrosis, *n* (%)	0	49 (36.0%)
	1	43 (31.6%)
	2	14 (10.3%)
	3	23 (16.9%)
	4	7 (5.1%)
NAFLD activity score, *n* (%)	not NASH (NAS ≤ 2)	30 (22.1%)
	Borderline NASH	19 (14.0%)
	Definite NASH (NAS ≥ 5)	87 (63.9%)

BMI: body mass index, AST: aspartate aminotransferase, ALT: alanine aminotransferase, NAFLD: non-alcoholic fatty liver disease, NASH: non-alcoholic steatohepatitis.

**Table 2 diagnostics-14-01083-t002:** Diagnostic performance of ARFI, APRI, FIB-4, and NFS for advanced fibrosis.

	AUROC (95% CI)	Cut Off	Sensitivity (%)	Specificity (%)	PPV (%)	NPV (%)	*p* Value
ARFI	0.832	>1.32	70.0	80.19	50.0	90.4	0.000
	(0.751–0.914)						
APRI	0.794	>0.637	73.3	76.4	46.8	91.0	0.000
	(0.711–0.878)						
FIB-4	0.767	>1.75	63.3	85.9	55.9	89.2	0.000
	(0.656–0.877)						
NFS	0.696	>−0.3	60.0	82.1	48.6	87.9	0.001
	(0.579–0.814)						

ARFI: acoustic radiation force impulse, APRI: aspartate aminotransferase-to-platelet ratio index, FIB-4: fibrosis-4 index, NFS: non-alcoholic fatty liver disease fibrosis score, PPV: positive predictive value, NPV: negative predictive value.

## Data Availability

The corresponding author or the first author can provide the data upon request.

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
