# Peer review of "Assessing the Utility of Acoustic Radiation Force Impulse in the Evaluation of Non-Alcoholic Fatty Liver Disease with Severe Obesity or Steatosis"

_diagnostics, 2024, doi:10.3390/diagnostics14111083_

Round 1

Reviewer 1 Report

Comments and Suggestions for Authors

It is an interesting paper, important for the readers of the journal.

Please add the role of liver MRI elastography to the introduction.

What was the exclusion criteria of the study? Please add this information to the text.

Was there any patient who had any focal liver lesion during the US scans?

Please add at least 1 US image to the paper.

In the abstract, please highlight the the ARFI imaging is based on Ultrasound. 

Author Response

Thank you very much for your time and effort in reviewing this manuscript. Below, you will find detailed responses and the corresponding revisions/corrections highlighted/in track changes in the re-submitted files.

Reviewer 2 Report

Comments and Suggestions for Authors

This article is well written but it adds no important new data to the readers.

Major points

1)             I cannot understand the aim of this study. It is to compare various non-invasive tools in the diagnosis of NASH or to emphasize the diagnostic superiority of ARFI in this field?

2)             ARFI: Please compare strictly the two ARFI machines. Please present the results of liver stiffness of each machine separately.

3)             Results of ARFI: Please present intra-and inter-observer variability.

4)             References: There were no new papers of US elastography of NASH.

Minor points

1)    This article included many abbreviations. Please add a list of abbreviations. It will help readers understand the content more easily.

Author Response

(The authors gave the same response as above.)

Reviewer 3 Report

Comments and Suggestions for Authors

In the work “Usefulness of Acoustic Radiation Force Impulse (ARFI) in non-alcoholic fatty liver disease with severe obesity or steatosis”, authors evaluated the non-invasive method of diagnostic test using ARFI for screening of advanced fibrosis in NAFLD patients. And this could be used as an alternative test for diagnosing fibrosis in NAFLD patients. The authors used multivariate statistical analysis to predict the diagnostic value of the clinical data in this present study. The way the data presented is good and is well written manuscript. I suggest authors to address the following comments in the revised manuscript for considering further.

1.    Please provide a Figure to depict enrolment or participant flow diagram to enable reader’s understanding.

2.    Please provide references for APRI, FIB-4, and NFS calculations though it is general formula used to calculate these parameters.

3.    I would like to suggest authors to increase font size of text used in the graphs of Figures 1, 2.1, 2.2, and 2.3 to enable reader’s understanding.

4.    I would like to suggest authors to check and verify once again all the statistical data presented in-text and tables throughout the manuscript. The data and significance should match the context.

5.    As general comment, please check any typographical errors and general English grammar throughout the manuscript.

Author Response

(The authors gave the same response as above.)

Round 2

Reviewer 2 Report

Comments and Suggestions for Authors

Article revision is insufficient. The manuscript has not been revised, and is not acceptable for publication.

Author Response

We sincerely appreciate your time and effort in reviewing this manuscript. We've detailed our response below and highlighted the changes to the resubmitted file. Thank you so much.
